PLoS ONE RESEARCH ARTICLE

# Retrospective spatial analysis for African swine fever in endemic areas to assess interactions between susceptible host populations

Jaime Bosch[1]*, Jose A. Barasona[1], Estefanía Cadenas-Fernández[1], Cristina Jurado[1], Antonio Pintore[2], Daniele Denurra[2], Marcella Cherchi[2], Joaquín Vicente[3], Jose M. Sánchez-Vizcaíno[1]

1 VISAVET Health Surveillance Centre, Animal Health Department, Faculty of Veterinary, Complutense University of Madrid, Madrid, Spain, 2 Istituto Zooprofilattico Sperimentale della Sardegna, Sardinia, Italy, 3 Spanish Wildlife Research Institute (IREC) (CSIC-UCLM), Ciudad Real, Spain

☯ These authors contributed equally to this work.
* jbosch@ucm.es

**Data Availability Statement:** Data of free-ranging pigs and notifications of ASF in domestic pig and wild boar cannot be shared publicly because it is

## Abstract

African Swine Fever (ASF) is one of the most complex and significant diseases from a sanitary-economic perspective currently affecting the world's swine-farming industry. ASF has been endemic in Sardinia (Italy) since 1978, and several control and eradication programmes have met with limited success. In this traditional ASF endemic area, there are three susceptible host populations for this virus sharing the same habitat: wild boar, farmed domestic pigs and non-registered free-ranging pigs (known as "brado" animals). The main goal of this study was to determine and predict fine-scale spatial interactions of this multi-host system in relation to the epidemiology of ASF in the main endemic area of Sardinia, Montes-Orgosolo. To this end, simultaneous monitoring of GPS–GSM collared wild boar and free-ranging pigs sightings were performed to predict interaction indexes through latent selection difference functions with environmental, human and farming factors. Regarding epidemiological assessment, the spatial inter-specific interaction indexes obtained here were used to correlate ASF notifications in wild boar and domestic pig farms. Daily movement patterns, home ranges (between 120.7 and 2,622.8 ha) and resource selection of wild boar were obtained for the first time on the island. Overall, our prediction model showed the highest spatial interactions between wild boar and free-ranging pigs in areas close to pig farms. A spatially explicit model was obtained to map inter-specific interaction over the complete ASF-endemic area of the island. Our approach to monitoring interaction indexes may help explain the occurrence of ASF notifications in wild boar and domestic pigs on a fine-spatial scale. These results support the recent and effective eradication measures taken in Sardinia. In addition, this methodology could be extrapolated to apply in the current epidemiological scenarios of ASF in Eurasia, where exist multi-host systems involving free-ranging pigs and wild boar.

confidential data. Data are available from the Regional Veterinary Epidemiological Observatory-Istituto Zooprofilattico Sperimentale della Sardegna (EOVR-IZS) (Institutional Data Access / Ethics Committee) for researchers who meet the criteria for access to confidential data. We confirm that others will be able to access this data in the same way as the authors. We also confirm that the authors did not have any special access privileges that others would not.

**Funding:** The present work was financially supported by the European project ASFORCE (FP7 - KBBE.2012) and Spanish project RTA2015-00033-C02-02 Instituto Nacional de Investigación y Tecnología Agraria y Alimentaria (INIA). JB is supported by postdoctoral "Juan de la Cierva" contracts FJCI-2015-23643 from MINECO-UCM and IJCI-2017-33539 from MINECO-UCLM. EC-F and CJ are recipients of Spanish Government-funded PhD fellowships for the Training of Future Scholars (FPU) given by the Spanish Ministry of Education, Culture and Sports.

**Competing interests:** The authors declare that the research was conducted in the absence of any commercial or financial relationships that could be construed as a potential competing or conflict of interest.

## Introduction

Wild and domestic suids act as reservoirs and potential carriers of many infectious diseases that come into regular contact with livestock and human [1]. African swine fever (ASF) is a viral disease of suids, affecting both domestic and wild animals [2]. ASF is notifiable to the World Organisation for Animal Health (OIE) due to its tremendous sanitary and economic consequences [3]. When ASF virus (ASFV) enters a naïve population of domestic swine, it causes high lethality rates of up to 95–100% [4]. The lack of treatments or vaccines makes ASF control and eradication a challenge, such that rapid detection of outbreaks and establishment of strict sanitary measures are the most relevant tools for controlling ASF [5]. However, these measures are insufficient to control ASF when wildlife are largely affected, such as in the current epidemical situation in the European Union (EU) [6], where more than 90% of notifications are from wild boar [7], or in the island of Sardinia (Italy), where ASF has been endemic for four decades, in which wild boar and other free-ranging host are also involved in the maintenance of the disease [8]. This situation underscores the need to assess the implications of virus-hosts spatial relationships in the epidemiology of ASF. Therefore, information of wild boar is crucial to contribute to the knowledge on interaction with domestic animals and risk of transmission of ASF in the current Eurasian context. In pastoral systems or agroforestry areas, domestic and wild species share the same space and the same resources, generating potential risks of cross-transmission of pathogens [9, 10]. This risk is higher when species are taxonomically close, which is the case of wild boar, free-ranging pigs and domestic pigs. Thus, the extensive production systems in pastoral areas, like those of Sardinia, where exist these three susceptible host populations for ASFV sharing the same habitat, is an ideal exemplary for understanding the dynamics of shared diseases among swine populations. Therefore, the retrospective spatial information in the epidemiological scenario of ASF in Sardinia could be used to assess the risk factors associated with interactions areas between suids susceptible host populations; and also to use this understanding in the current scenarios of ASFV in Eurasia.

The island of Sardinia is the only region outside Africa that has remained ASF-infected since the previous introduction of the disease in the European continent in the 1960s [4]. In this sense, the island has been infected for more than 40 years, since 1978 [8], and it has become an endemic scenario where multi-host cycles are implicated: domestic farm pigs, wild boar and free-ranging pigs [11–13]. Many studies have shown that the persistence of the disease in Sardinia is related to lack of farm professionalism, high density of wild boar population and presence of non-registered pigs allowed to range freely (free-ranging pigs; known locally as "brado" animals) [14–18]. ASF endemicity on the island is not related to the *Ornithodoros* ticks that serve as a disease vector in other Mediterranean countries, since these ticks are absent from Sardinia [18]. Based on these facts, Sardinian authorities have promoted good farming practices in their ASF eradication plan, and they prohibited free-ranging pigs in 2012 [19]. However, this traditional animal breeding continues largely as a result of socio-cultural factors [14], and until this year there was no information available on sanitary status of these free-ranging pigs [8]. More than 50% of free-ranging pigs are seropositive for anti-ASFV antibodies and more than 2% were infected with the virus; this prevalence was much higher than that among farmed domestic pigs and wild boar, highlighting the key role of free-ranging pigs as a main source of ASFV in the island [8]. The first ASF eradication programme achieved its goal in southern Sardinia, but the disease has remained endemic in north, central and eastern regions. The central eastern part of the island is the most affected and is known as the traditionally endemic ASF area in Sardinia [8]. This region is characterised by mountainous areas, where free-ranging pig farming is especially common [12, 13], and it is where the abundance of free ranging pigs and the ASF seroprevalence are highest [8]. Previous studies have

identified the combination of mountainous areas with high density of wild boar as a significant risk factor for ASF occurrence [15]; thus, wild boar and free-ranging pig interactions may act as an important driver of ASFV for farmed domestic pigs.

During the past 15 years, ASF has been actively moving through Africa, increasing the presence of ASFV and reaching a historical record of affected countries, many of them previously free of the disease [20]. The primarily causes of these ASFV spread were the increase in pig production and the globalization of communications within Africa and from Africa abroad [21]. Since 2007 till 2014, ASF has spread throughout the Caucasus region, affecting neighbouring countries such as the Russian Federation and Belarus, and later reaching countries of the EU. In the north and south of the Russian Federation, pig farms of low biosecurity have been described as the main driver of the ASFV [22, 23]. In addition, the wild boar has played an active role in ASF epidemic in Europe, being involved in the introduction and local dissemination within the eastern EU countries. However, although in these endemic countries the wild boar has played a secondary role in the transmission of ASF [23, 24], this wild ungulate can transmit the virus, even in the absence of domestic pigs [11]. Thus, in the last seven years, many research have been developed to help to contain and to prevent the dispersion of the disease through creation of models and simulation, such as: the assessment of the risk of ASF introduction into the EU by wild boar [25] and into disease-free EU countries [26]; the prediction of the global ASF outbreaks [27]; the analysis of the distribution and dispersion of the ASF in Poland [28]; the impact of forestry and leisure activities on wild boar with the risk of ASF spread [29]; the identification of the role of wild boar in the spread of ASF in Russia [30]; the assessment of ASF introduction to Japan via pork products brought in air passengers' luggage [31]; the assessment of the risk of ASF in the south-eastern countries of Europe [32] and even the analysis of the geographical expansion of wild boar in Eurasia and their biological cycle for more effective decision-making about health and natural resource management [33].

Currently, ASF is present in 29 Eurasian countries, affecting both domestic pig populations and wild boar populations. In Europe it affects 19 countries (Georgia, Armenia, the Russian Federation, Azerbaijan, Ukraine, Belarus, Lithuania, Latvia, Estonia, Poland, Moldova, Czech Republic, Belgium, Bulgaria, Hungary, Romania, Slovakia, Serbia and Greece) and 12 countries in Asia (China, the Asian part of Russia, Mongolia, Vietnam, Cambodia, North Korea, South Korea, Laos, Myanmar, Philippines, East Timor and Indonesia), where it spreads more quickly and unstoppably [7]. However, despite the prevention and control measures carried out, ASFV continues its geographical progress through the affected countries [26, 34, 35]. The situation of ASF in Sardinia shares many characteristics with some of the epidemiological scenarios that have been generated and which have evolved in Eurasia. These scenarios will vary depending on the role of wild and domestic hosts, the importance of environmental, the type of exploitation and the level of biosecurity of the country's farms, commerce, social and cultural factors or traditions of the country and the susceptible host population involved [4]. Consequently, currently in the countries of Eastern Europe, in the EU, Asia and Africa, the transmission and maintenance of ASF is occurred under different epidemiological scenarios, involving both domestic pigs and wild boars [4, 9]. Therefore, knowing the areas of interaction between the suids host populations will be essential to help design and implementing risk-based interventions for the prevention, control, surveillance and early detection of ASF; main methods and tools to fight against this disease.

Understanding the spatial interactions between free-ranging pigs and wild boar may aid in efforts to control and eradicate the disease in Sardinia scenario of ASF, where these two host play an important role and could potentially favour the transmission of ASFV to farmed domestic. The present study took advantage of advances in global positioning system (GPS) technology [36–38] to determine and predicted fine-scale spatial interactions between wild

boar and free-ranging pigs in the main endemic area of Sardinia. Extensive production systems in pastoral areas, like those of Sardinia, are therefore an ideal exemplary to study and develop a methodology to identify the spatial interactions areas between suids susceptible host populations and understanding the risk factors associated. However, a particular purpose of this retrospective preliminary study is to be able to extrapolate this very innovative methodology in other epidemiological scenarios of ASF in the current Eurasian context. Particularly to apply in epidemiological scenarios of ASF where exist potential risks of ASFV transmission to other free-ranging pigs, areas with low biosecurity pigs farms and especially where wild boar play an important role in this disease.

## Materials and methods

### Study area

The island of Sardinia (Italy) is located in the center of the western Mediterranean Sea, with dry and hot summers and rainy, mild winters. The climate is typically Mediterranean: mean annual temperature ranges from 11.6˚C to 18.0˚C; and annual precipitation, from 200 to 500 mm [39]. The island shows a complex orographic pattern and a wide variety of biotopes. The most extensive vegetation cover on the island are Mediterranean shrubs, maquis, garrigue and broadleaf forests of mainly *Quercus ilex*, *Q. suber* and *Q. pubescens*, combined with herbaceous pastures and mixed agricultural areas [32, 33]. We conducted the study in the traditionally ASF-endemic area of Montes-Orgosolo, between the Nuoro and Ogliastra regions, in the central eastern part of the island (Fig 1). Here rates of ASFV infection in free-ranging pigs and wild boar are persistently high [8, 12, 13, 15]. The study area comprises a natural protected reserve and nearby pastoral area that domestic pigs, wild boar and free-ranging pigs co-inhabit [40]. The study site lies within the Orgosolo Municipality, which has been defined as the ASF epicentre in Sardinia [8].

The interaction between wild boar and free-ranging pig denoted in this study as spatial interspecific interaction (SII) index, was quantified in two steps. In the first step, we analysed the wild boar and free-range pig data collection and the environmental factors to obtain the spatial information used in the model. It is recalled that this study it only based on to identify the interaction between wild boar and free-ranging pig and their potential ASFV transmission between them or to domestic pig farms, without taking into account others drivers for ASFV infection on farmed domestic pigs (mainly animal movements). In the second step, the SII index at fine scale, was calculated in the study area based on wild boar and free-ranging pigs telemetry and occurrences data and the environmental factors that explain occurrences of both suids populations. Finally, we assessed of the relative interaction index map as a tool for ASF epidemiology, overlapping the ASF notifications of wild boar and domestic pigs onto predicted SII values.

### Wild boar and free-range pig data collection

We used data from telemetry technology to track adults wild boar (males 7000, 7001 and female 7003) from the area of Montes-Orgosolo that were equipped with a GPS-general packet radio services (GPRS) radiocollar (Microsensory System, Spain) and proximity contact loggers. Wild boar capture followed a protocol designed and developed by scientists (categories B and C of animal experimentation) in accordance with EC Directive 86/609/EEC for animal handling and experiments, as part of a wild-domestic suids spatial exploring disease ecology study (http://www.asforce.org/). We captured wild boar from different social groups using padded foothold cage traps monitored using camera traps, as described by Barasona et al. [41]. Radiocollars were programmed to acquire one fixed GPS location per hour for seven days a week between March and September 2014. GPS locations were associated with a mean

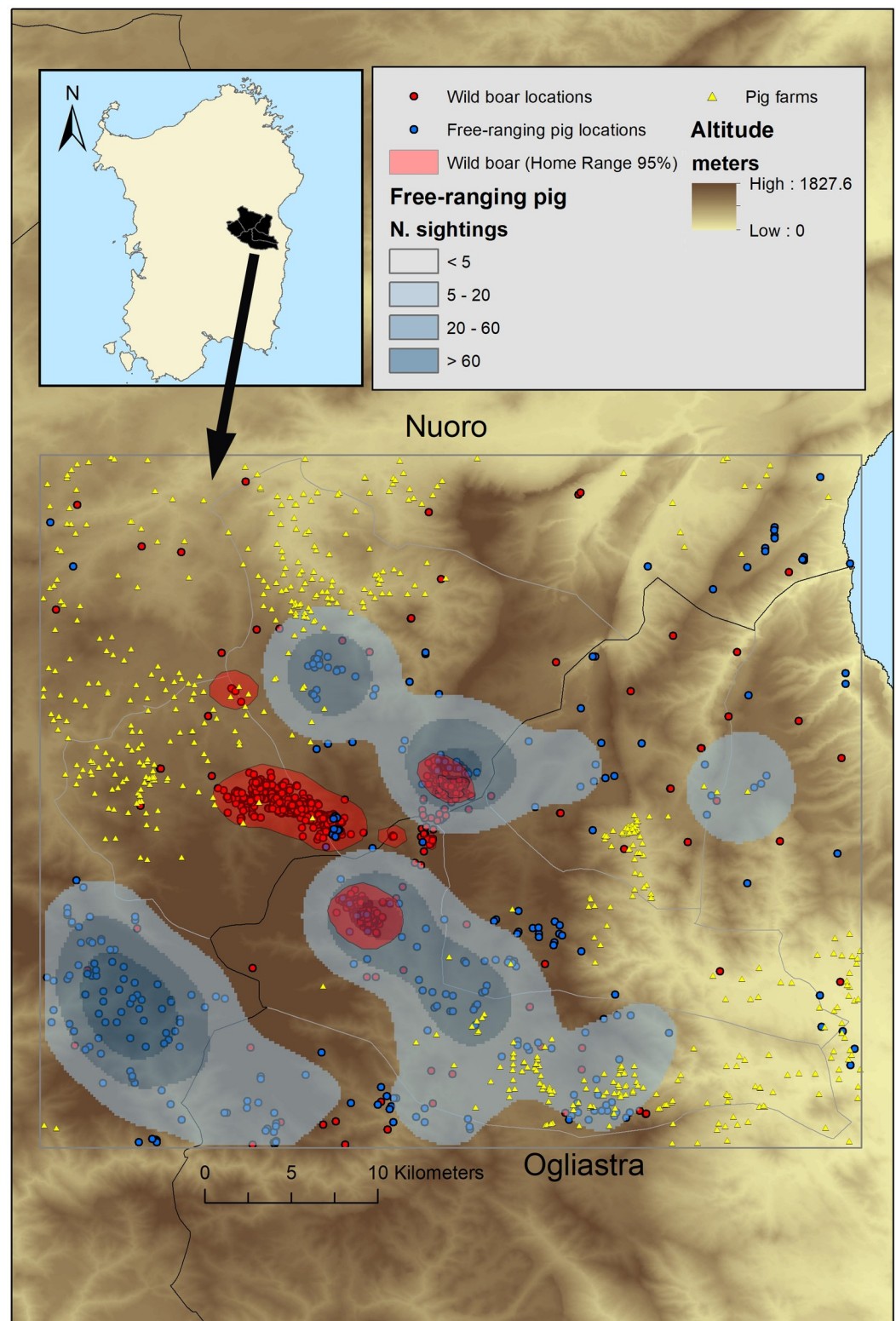

**Fig 1. Study area, Home Range (HR) of wild boar, and spatial distribution of free-ranging pigs and pig farms in Montes-Orgosolo.**

positional error of 26.6 m (SD = 23.5 m) following the strategy of Barasona et al. [42]. Furthermore, we used additional spatial information about the occurrence records of wild boar [9] in order to complement and to prove the presence/absence of wild boar in the study area.

We obtained occurrence records of free-range pig sightings from 2012 to December 2015 provided by the Regional Veterinary Epidemiological Observatory-Istituto Zooprofilattico Sperimentale della Sardegna (EOVR-IZS), in that legally prohibited from collaring free-ranging pigs. These data are based on population monitoring of free-ranging pigs and their carcasses in the field and depopulation actions against free-ranging pigs. All these actions were coordinated by the veterinarian responsible for the ASF eradication at the Animal Health Services of the Local Health Authority of Sardinia, the forest guards of Nuoro and highly specialised task force in eradication diseases activities. The main tasks of these groups were to sight free-ranging pigs, record the number and geographical location of animals spotted and prepare for the subsequent depopulation actions. We considered the population monitoring of free-ranging pig sightings and the wild boar GPS recording, including the occurrence records, as a single spatio temporal window for analysis.

## Environmental factors

Data on climatic factors were obtained from WorldClim [43]. Data on elevation topography were obtained from the Global Land Cover Facility (GLCF), and slope and topographic diversity were generated from elevation data. Topographic diversity represents the topographic complexity of the terrain, i.e. the sum of the different number of slopes, elevations, and orientations around a given cell. Topographic position was defined as the difference in magnification between a cell and the surrounding cells in a given radius, and data on this factor allowed the differentiation of high, flat or depressed areas. We applied here the categories of quality of available habitat (QAH) described in our previous work [9]. Data on solar radiation were obtained from the Shuttle Radar Topography Mission (http://www.glcf.umd.edu/data/srtm); data on quantity and quality of vegetation development, from the Normalized Difference Vegetation Index datasets of the Global Inventory Monitoring and Modeling Studies [44]; and data on percentages of bare, herbaceous, or tree cover, from the Vegetation Continuous Fields (http://www.glcf.umd.edu/data/gimms). This last dataset captures variations in the vegetation index during the 12 months of 2014. This index can range from -1 to +1; a zero means no vegetation and values of 0.7–0.9 indicate the highest possible density of green leaves. Anthropogenic or human influence was approximated using the human footprint raster [45] obtained from the Socioeconomic Data and Applications Center (http://sedac.ciesin.columbia.edu/data/collection/wildareas-v2/sets/browse). Distances to inland water (rivers, canals, and lakes) were obtained from the Digital Chart of the World of the US National Geospatial-Intelligence Agency. Distances to farms were obtained from EOVR-IZS.

In order to reduce collinearity among the environmental factors, they were entered into a correlation matrix, and redundant factors were identified based on distance (shorter distance = higher correlation) using Pearson correlation coefficient and the Hierarchical Clustering through the raster and cluster R-package [46, 47]. Factors were selected based on a cut-off or threshold minimum of 0.5. One factor was selected from each group with node < 0.5 [48]. Since factors surviving this selection may themselves be linear combinations of other factors, we calculated the variance inflation factor by sequentially removing factors with higher values (maximum value allowed = 5) [49].

## Spatial overlap among suid occurrences

We estimated home-range (HR), corresponding to the 95% utilization distribution (UD) [48], using data from radiocollared wild boar and the fixed-kernel function in the ADEHABITAT package [50] in R version 3.6.0 (R Development Core Team). To evaluate interspecific patterns of space use in the heterogeneous study environment, fixed-kernel density estimators were applied, allowing the identification of disjunctive areas of activity [48].

Using the GPS locations of wild boar, we determined the maximum distance travelled by wild boar in in the study area, daily average distance travelled and the pattern of activity. We also used GPS locations to model the spatial distribution of free-ranging pigs and occurrence locations of wild boar in the study area using kernel density estimations and a 26-m radius (based on GPS positional error). Kernel density estimation converts intensities of free-ranging pig locations to abundance gradient, producing a smooth density surface of distribution over the study area [25].

The surface distribution of HRs and occurrence of wild boar and the surface distribution of free-ranging pigs were used to estimate spatial overlap between wild boar and free-ranging pigs [51] within the study area (Fig 1). The extent of spatial overlap between wild boar and free-ranging pigs was calculated as the area of wild boar HR overlapping with the polygonal distribution contour of free-ranging pigs, divided by the total wild boar HR. Analogously, the extent of spatial overlap between wild boar was calculated as the area of HR overlapping for the two different wild boar, divided by the total HR of both wild boar.

## Spatial inter-specific interaction index at fine scale

The extent of overlap between wild boar HR and spatial distribution of free-ranging pigs is a relatively coarse indicator of potential interactions, as it is based on distributions [51]. Following the approach of Barasona et al. [42], we applied latent selection difference (LSD) functions [52, 53] to assess fine-scale interactions between wild boar and free-ranging pigs, as well as determine differences in the environmental factors evaluated. This analysis determined which covariates predicted similarities and differences in resource use between species at smaller spatial scales, measured by the estimated β coefficients from logistic regression [54–56]. Significant coefficients indicate less or more use by one species compared to other one [57, 58]. Positions of free-ranging pigs and wild boar were transformed into buffer areas of radius 26 m (to account for GPS positional error), and within each buffer we calculated the selected environmental factors (predictors) after they were reduced to avoid collinearity. These factors included straight-line distance (km) to water resources; straight-line distance (km) to farms; proportion of total cover that was bare, herb or tree; proportional cover of wild boar QAH; solar radiation; human influence; and other factors related to climate, topography and vegetation. These factors were selected because they are usually used to model wild boar distribution, and they are highly likely to influence interactions between wild boar and free-ranging pigs in the study area [9, 15, 9, 45, 59, 60].

To compare resource selection of wild boar and free-ranging pig, we used logistic regression to estimate coefficients for LSD functions in the RMS package routine [61] in R version 3.6.0 (R Development Core Team R, 2018). For this analysis, we coded locations from wild boar as 0 and from free-ranging pigs as 1. Factors associated with significant positive coefficients were preferred to a greater extent by free-ranging pigs than wild boar; those associated with significant negative coefficients were avoided to a greater extent by free-ranging pigs than wild boar. The distance to factors should be interpreted the opposite way, such that those with non-significant coefficients indicate habitats with the highest potential for interspecific interactions, because the two species do not differ in their selection of these resources. The main

assumption of LSDs is that all resources should be equally available to both species within the study area. The selected study area involves genuine inter-species contact or overlap contours (as cited in Latham et al., 2011).

In order to validate the outcome model, we randomly split the datasets using 70% of locations to parameterise the models (training datasets) and the remaining 30% of locations for model validation (independent, validation datasets) [62]. The best model was obtained using a forward–backward stepwise procedure on the training datasets based on Akaike Information Criteria [63], allowing us to obtain the most parsimonious model. We assessed predictive capacity of the best model using calibration plots: the best models were tested against the corresponding validation dataset, then the observed and predicted frequencies of observations were plotted for 10 equally sized intervals of predicted probabilities (0–1). A model with high predictive capacity should show perfectly aligned points along a 45° line [64]. We also assessed the ability of the model to discriminate free-ranging pig and wild boar locations by determining the area under the receiver operating characteristic curve (AUC). The best possible AUC is 1 (models with perfect discrimination ability), while a value of 0.5 suggests that the model performs no better than random [64].

The best LSD model was used to spatially map the relative probability of use (P) by free-ranging pigs relative to wild boar. Areas with P of approximately 0.5 were further analysed, since they feature the highest probability of spatial interactions between the two species [52]. Accordingly, we used the spatial interspecific interaction (SII) index described in Barasona et al. [42], (i.e. SII = (1 –P) if P $\geq$ 0.5, and SII = P if P < 0.5). Further, we used a model trained with data from the study area to extrapolate P and a derived SII index for 600-m cells across a wider region than the study area.

To assessment of the relative interaction index map as a tool for ASF epidemiology, ASF notifications in wild boar and domestic pigs from 2010 to 2016, based on data from the EOV-R-IZS, were mapped onto predicted SII values. The notifications included cases on pig farms as well as serology and virology laboratory results obtained from case reports and governmental agencies. To facilitate interpretation, the predicted SII values were converted into a low interaction of 1 (25th percentile), medium interaction of 2 (50th percentile), or high interaction of 3 (75th percentile). The frequencies of notifications within each category were calculated. In addition, we also considered distances closer than 300 m from ASF notifications mapped to predicted SII values for taking in to account the indirect transmission generated at the local spread. We used a moderate selection distance, since the indirect transmission of ASF was the most common route of transmission of secondary cases between farms within a 2 km radius through fomites, being the most significant risk factors the high densities of free-ranging pig and wild boar [13]. ANOVA tests were then performed to examine whether there were significant differences in the spatial occurrence of ASF notifications in wild boar and in domestic pig respect to SII local indexes. In the north central zone of Sardinia, ASF notifications in domestic pigs were compared with ASF-negative domestic pig farms in terms of SII indexes. ASF notifications in wild boar were also compared with a random data set in terms of SII indexes. All statistical analyses were performed in R version 3.6.0.

## Results

In this study, we analysed four years of sampling data for free-ranging pigs (5401 locations, 2012–2015), occurrence records of wild boar (1133 locations, 2011–2015) and real movement data from wild boar with more than 5150 hours of data along 248 days (4288 locations, 2014). The locations of free-ranging pig sightings and wild boar were distributed widely, both species showed a similar spatial distribution over time in the study area. Collared wild boar were

distributed across the Montes-Orgosolo between the Nuoro and Ogliastra regions, tending to occupy the central part of the study area (Fig 1). Wild boar were also observed in more remote areas.

Analysis of the correlation tree (see S1 Appendix) and the variance-inflated factors ($< 5$) reduced the initially factors to 15 in the LSD approach (see S2 Appendix). These factors were grouped as climatic (n = 2), human influence *(2)*, topography *(5)*, and vegetation *(6)*. The two male wild boar 7000 and 7001 remained initially close to the capture site, then they established respective HRs of 9.3 and 9.2 km from the capture site (Table 1). Female 7003 remained near the capture site throughout the study period and established an HR of only 1.2 km, often entering the area where free-ranging pigs were present. Her HR overlapped with that of male wild boar 7001. Wild boar are a crepuscular/nocturnal species, and their activity peaks typically between 23:00 h and 05:00 h. In the study area, wild boar were most active between 18:00 h and 05:00 h (see S3 Appendix), showing average movement of 348.7 meters per hour. The average distance moved by male wild boar increased in March, August and September. In June and July, the female wild boar moved more than male 7000.

Estimated distribution contour was substantially larger for free-ranging pigs (9,744.37 ha) than for wild boar (4,235.89 ha) (Fig 1). Overlap between both distribution contours was 66.2% overall, 91% for wild boar 7001, 7.01% for boar 7000 and 99.6% for boar 7003. The estimated average HR (Kernel, 95% UD) in wild boar was 1411.9 ha. The two adult male boar 7000 and 7001 remained initially close to the capture area, and after that established respective HRs of 9.3 km and 9.2 km from the capture site (Table 1). Overlap between the HRs of wild boar 7001 and 7003, which came from different family groups, was 8.05%. The signal of the GPS transmitter of wild boar 7003 was not continuous during the entire study period. We assume that its HR overlapped more with that of wild boar 7001 than with that of wild boar 7000.

Fine assessment of spatial interactions between wild boar and free-ranging pigs revealed that the environmental variables that appeared to explain habitat selection differed between the two species (Table 2). Free-ranging pigs used bare areas, tree areas such as broadleaved deciduous forest, and areas with higher annual temperature ranges to a significantly greater extent than wild boar. Free-ranging pigs showed weaker preference than wild boar for areas with higher elevation, areas with higher proportions of slopes, areas with higher proportions of geographic position (areas with higher proportions of elevated or depressed areas), areas with more heterogeneous areas of precipitation seasonality, and areas with smaller averaged

**Table 1. Distances travelled daily by wild boar (meters), the total number of GPS locations, and the number of locations monitored by GPS-GSM for consecutive hours.**

| | All locations | | | Locations monitored for consecutive hours | | | |
|---|---|---|---|---|---|---|---|
| | 7000 (♂) | 7001 (♂) | 7003 (♀) | 7000 (♂) | 7001 (♂) | 7003 (♀) | X̄ |
| **n** | 1,267 | 2,742 | 279 | 1,025 | 2,429 | 239 | 1,231 |
| **Min** | 0 | 1 | 1.4 | 1 | 1 | 1.4 | 1 |
| **Max** | 12,959.4 | 9,217.9 | 1,241.1 | 9,327.2 | 9,217.9 | 1,241.1 | 6,595 |
| **X̄** | 1,015.5 | 148.8 | 139.1 | 770.0 | 144.3 | 132.0 | 349 |
| **Me** | 194.6 | 49.6 | 69.6 | 184.6 | 52.8 | 69.1 | 102 |
| **SD** | 2,155.4 | 292.2 | 183.3 | 1,760.6 | 287.7 | 179.1 | 742 |
| **SE** | 60.55 | 5.58 | 10.97 | 54.99 | 5.84 | 11.58 | 24 |

Abbreviations: n, number of samples; min, minimum travelled distance (meters); max, travelled distance (m); X, mean travelled distance (m); SD, standard deviation; SE, standard error.

**Table 2. Latent selection difference model to identify environmental factors that explain habitat selection by wild boar and free-ranging pigs.**

|  | LSD model | |
|---|---|---|
|  | (AUC = 0.906) | |
|  | β | SE |
| (Intercept) | -29.70*** | 7.95 |
| Topographic diversity | 0.57*** | 0.03 |
| Elevation | -0.01*** | 0.00 |
| Precipitation Seasonality | -1.04*** | 0.09 |
| Temperature Annual Range | 3.59*** | 0.30 |
| Topographic position | -0.01*** | 0.00 |
| Normalized Difference Vegetation Index averaged | -8.66*** | 1.20 |
| Slope | -0.13*** | 0.02 |
| Percentage of tree | 0.02*** | 0.00 |
| Percentage of bare | 0.08*** | 0.02 |
| Distance to pig farms | 0.04. | 0.03 |

(P-values:. = 0.1, * = p < 0.05, ** = p < 0.01, *** = p < 0.001).

The table shows model coefficients (β), standard errors (SE) and area under receiver operating characteristic curve (AUC) from LSD functions used for determining relevant factors explaining differences in habitat use by wild boar (coded as 0) and free-ranging pig (coded as 1) in Sardinia. Variable names are described in section 2.3.

normalized difference vegetation index (NDVI) (Table 2). This last index measures the quantity, quality and development of vegetation.

Despite these substantial differences in environmental factors, our prediction model showed the highest spatial interactions between both hosts in the areas close to pig farms, since non-significant relation was found in straight-line distance (km) to farms (Table 2). The LSD model showed good discriminatory power (AUC = 0.906) and predictive reliability (see S4 Appendix), supporting its use in extrapolating spatial patterns of SII across the Montes-Orgosolo as well as in central and northern regions of the island (Fig 2).

The SII index as a tool for ASF epidemiology were assessed obtaining: of the 453 ASF notifications, 28.9% lay within the area of maximum interaction (level 3) between free-ranging pigs and wild boar (Table 3; Fig 2); 26.4% and 33.1% of notifications corresponded, respectively, to domestic pigs and wild boar (Table 3). However, these same three percentages rose to 62.7%, 60.3% and 66.3% when we considered only distances closer than 300 m from ASF notifications (Table 3). ASF notifications in domestic pigs occurred significantly more often in areas with high predicted SII values (ANOVA test analysis; $F = 30.85$; $p<0.01$), as did ASF notifications in wild boar (ANOVA test analysis; $F = 46.14$; $p<0.01$).

## Discussion

This study assessed spatial interactions between wild boar and free-ranging pigs in a region where ASF has remained endemic for the longest time outside Africa. The goal was to understand how resource selection preferences, and the resulting spatial overlap, from these species, may contribute to ASFV transmission and persistence. This is, to the best of our knowledge, the first fine spatial analysis aimed at explaining the patterns of ASFV transmission in multi-host system in Sardinia. Therefore, our research contributes to a better understanding of how ASFV hosts interaction are established helping to design interventions targeting the interaction. In this sense, our model may aid in surveillance programs as well as depopulation efforts

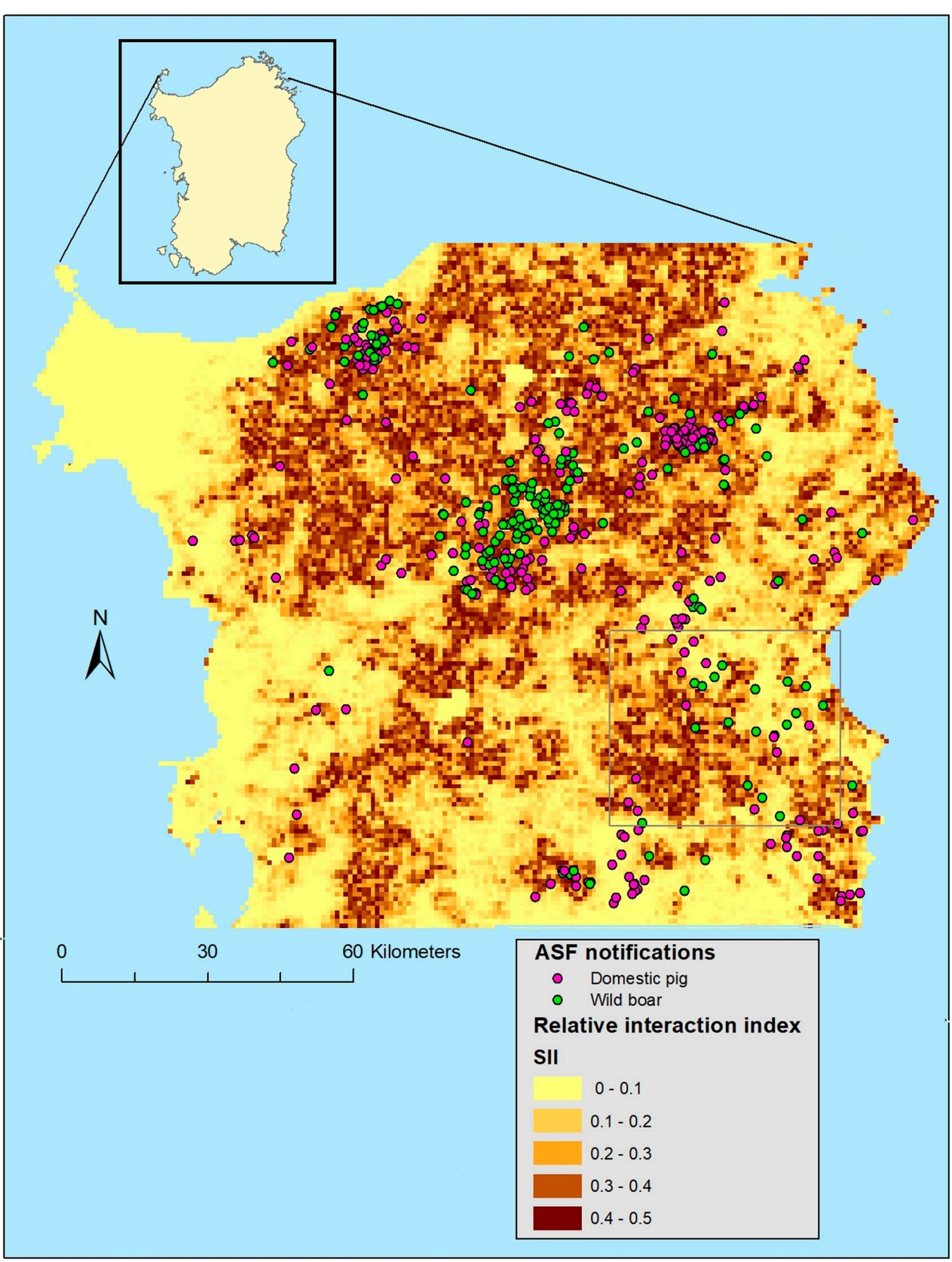

**Fig 2. Spatial interspecific interaction prediction derived from a latent selection difference model to identify environmental factors that explain habitat selection by wild boar and free-ranging pigs in the ASF-endemic area of Sardinia; and ASF notifications spatial overlapped with areas of different relative interspecific interaction values.**

against free-ranging pigs, which have the highest ASFV prevalence and seroprevalence of the three susceptible populations of the island [8]. In addition, this methodology it can be to apply in other epidemiological scenarios of ASF in the current Eurasian context. Especially in epidemiological scenarios where could exist potential risks of transmission of ASFV between free-ranging pigs, areas with low biosecurity pigs farms and wild boar populations.

The HR, daily distance travelled and resource selection patterns of radio-collared Sardinian wild boar are similar to results from radio-collared wild boar in other Mediterranean ecosystems [42, 65]. Nevertheless, our results from wild boar showed slightly higher diurnal activity compared with others studies in Mediterranean ecosystems (see S3 Appendix), except for a study on the south of Spain [42, 66], where some sporadic diurnal activity of wild boar was observed. However, our results should be verified and extended in studies collecting data for more time, preferably from different wild boar populations throughout the island. It had to be emphasized that is important to have obtained for the first time radio-collared three wild boars on the island of Sardinia. This is largely due to that was difficult obtaining field data from wild species in a complex territory like Sardinia that presents many logistical problems at local scale, mainly mediated by the AFSV problem. This region has multiple difficulties regarding conflicts among farmers, hunters, game-managers, scientists and authorities. Nevertheless, we obtained the first ranges of homes, distances travelled and spatial behaviour in one of the oldest endemic area of ASF in Europe. Knowledge of dispersal distances of wild boar on the island will be helpful for designing monitoring programmes and control measures, since these animals can affect the rate of disease spread and probability of new outbreaks [25, 67]. Adult males have a different behaviour and spatial utilisation than social female groups. Consequently, in our study, the analysis of adult males, is more interesting than females. This is because males covering greater dispersion distance than females [68, 69], and therefore with a greater probability of direct or indirect spatial interactions between the three swine susceptible populations of the island. In addition, the records of movements and preferred habitat of wild boar could be helpful for selecting areas to disperse baits in the field for future oral vaccine against ASF [70]. This aims to eradicate the disease in wildlife as it has been demonstrated in the past successful experiences with other pathogens, such as the oral vaccination against classical swine fever on wild boar in Germany [71].

We found in the study area that free-ranging pigs had larger spatial distribution than wild boar. In part, this likely reflects the raising of free-ranging pigs in "communal lands" within Montes-Orgosolo: these habitats are determined by humans, rather than freely chosen by the animals [8, 15, 72]. Mean HR for wild boar was somewhat smaller in our study area than in

**Table 3. Overlap between ASF notifications and areas of low, medium or high relative interaction between wild boar and free-ranging pigs.**

| | No distance cut-off | | | | | | Less than 300 m from coordinate of ASF notification | | | | | |
|---|---|---|---|---|---|---|---|---|---|---|---|---|
| | Domestic pig | | Wild boar | | Both | | Domestic pig | | Wild boar | | Both | |
| SII category | n | % | n | % | n | % | n | % | n | % | n | % |
| Low (1) | 103 | 37.1 | 64 | 36.3 | 167 | 36.9 | 62 | 22.4 | 39 | 22.1 | 101 | 22.3 |
| Medium (2) | 101 | 36.5 | 54 | 30.6 | 155 | 34.2 | 48 | 17.3 | 20 | 11.4 | 68 | 15.0 |
| High (3) | 73 | 26.4 | 58 | 33.1 | 131 | 28.9 | 167 | 60.3 | 117 | 66.5 | 284 | 62.7 |
| Total | 277 | 100 | 176 | 100 | 453 | 100 | 277 | 100 | 176 | 100 | 453 | 100 |

other Mediterranean areas [65, 67, 73]. This result should be verified based on a complete 12-month period of data collection. Despite this difference in HR, the spatial distribution of free-ranging pigs and wild boar overlapped by 66.2% overall. In fact, the overlap was 91% in the case of wild boar 7001 and 99.9% in the case of 7003, although only 7.01% in the case of wild boar 7000. In general, free-ranging pigs were frequently detected in areas used intensively by wild boar. Despite we compared their HR indirectly, due to the use of different methodology to determine it, these results are in line with other studies that assert that free-ranging pigs share the same habitat as the wild boar [8, 15, 40, 74]. This overlap likely reflects that both species prefer heterogeneous habitats offering shelter and natural food such as acorns, chestnuts and hazelnuts [40, 74, 75]. Also, we found that free-ranging pigs and wild boar tend to concurrently use areas close to pig farms, this could be explained by the attraction of animals to human-managed areas by the availability of food, shelter and water, facilitating the spread of ASFV and hindering its control.

Our modelling suggests that free-ranging pigs in the study area select habitat based on greater topographic diversity of the terrain, greater tree cover, lower bare soil coverage and narrower annual temperature range. These factors are not typical of models explaining habitat selection by free-ranging pigs in other Mediterranean ecosystems [37]. We suggest that this situation may reflect depopulation actions that were occurring in parallel with collection of data on sightings of free-ranging pigs [8]. Such actions may have driven the animals to avoid hunters by moving to areas with more tree cover and more diverse topographic terrain. The fact that free-ranging pigs are a species of human management means that they tend to inhabit areas with narrower temperature range than wild boar. Conversely, our results suggest that wild boar in the study area select habitats with higher elevation, higher proportion of slopes, higher proportion of geographic position, higher NDVI and a lower precipitation seasonality. These factors have also been shown to influence wild boar distribution in similar ecosystems [46, 59, 76].

Regarding ASF epidemiology, we found significant association between ASF notification in the north central area of the island, from wild boar and farmed domestic pigs, and the predicted SII indexes at fine-spatial scale in the study area. Therefore, we can conclude that the methodological approach to predict spatially explicit SII indexes carried out in this study it could help explain, in part, the risk of ASF presence. In this sense, the interaction between wild boar and free-ranging pigs represents a relevant risk for ASF presence, not only in wild boar but also in farmed domestic pigs. Interestingly, we observed that 43.2% of ASF notifications involving wild boar occurred in areas they preferred, while only 22.2% occurred in areas preferred by free-ranging pigs. This suggests that intraspecific contact among animals from different family groups could be also relevant to explain the transmission of the disease, this fact has already been shown in other regions of Europe [77].

In the current study, we used LSD modelling to estimate similarities or differences in the use of shared resources [42], which may help assess potential routes of direct or indirect ASF transmission. Unfortunately, since our data for free-ranging pigs that were collected in 2014 were less than wild boar, we were unable to examine whether the two species overlapped in time, preventing us from concluding absolutely whether the temporal overlap was associated with transmission. However, as other studies assert [8, 15, 40, 74, 75], both suids populations are species with similar spatial habits year after year, being in line with our research where exist in other years the spatio-temporarily overlap. In line with our previous findings, other study in the same area through camera trapping revealed, that the free-ranging pigs indirectly interacted often with wild boar and they also directly interacted, being these interspecific interactions higher than other observed in Mediterranean scenarios [78]. Despite this limitation, spatial predictions from the LSD approach can be extrapolated to a larger area where they can

be related to spatial risk of interspecific disease transmission [42]. Future studies should examine telemetry data over at least one year for the three swine susceptible populations of the island, or at least for wild boar and farmed domestic pig [8]. Another additional factor to take account when modeling the inter-specific interaction is the population abundance of free-ranging pigs and wild boar. However, we could not consider this factor in our study due to the lack abundance data these suids population at fine-spatial scale, but it would be greatly recommended for further studies. Forthcoming research could also combine proximity loggers and GPS technology to validate rates of interspecific contact, quantify the potential for indirect disease transmission, and identify habitats where both these events occur most frequently.

Finally, based on the results obtained from this study, we can conclude that the inter-specific interaction between free-ranging pigs and wild boar represents an important risk factor to explain the relict ASF Sardinian endemism in farmed domestic pigs, as previously suggested [8, 15]. Our results support the idea that raising free-ranging pigs in communal lands has strongly contributed to ASF persistence in Sardinia [8, 15, 12, 14, 79]. Also, our model provides the resource selection preferences, and the higher interaction areas between free-ranging pigs and wild boar. These observations mean a relevant step to clarify some of uncertainties points related to the epidemiology of ASF and are useful to establishment control and eradication measures of the disease focus on the interaction among multi-hosts, such as oral vaccination strategy [71], population control by selective culling and habitat management. It could be even used to improve the biosecurity on domestic pig farms in areas with high predicted SII values. Such integrated approaches have worked well against tuberculosis, Aujeszky's disease and classical swine fever [71, 80, 81]. On the other hand, it is a good chance to bet for this methodology and apply in other epidemiological scenarios of ASF in Eurasia where a great need to control this disease exists. In particular, to implement it in areas or regions with potential risks of transmission of ASFV between free-ranging pigs, areas with low biosecurity pigs farms and wild boar populations.

## Supporting information

**S1 Appendix. Cluster dendrogram or correlation matrix of the environmental variables included as predictors in the latent selection difference function (LSD) approach.** For definitions of variables see section 2.3 (material and methods).
(DOCX)

**S2 Appendix. Environmental variables included as predictors in the latent selection difference function (LSD) approach.** For definitions of variables see section 2.3 (material and methods).
(DOCX)

**S3 Appendix.** Average distance (in meters) per hour (a) and average distance (meters) per month (b) for wild boar number 7000 (adult male), 7001 (adult male) and 7003 (adult female) for the period from March 2014 to September 2014.
(DOCX)

**S4 Appendix. Calibration plots of the predictive performance of the best latent selection difference model (Table 2) to identify environmental variables that explain habitat selection by wild boar and free-ranging pigs.** The observed frequencies of free-ranging pig locations in the validation dataset are plotted as a function of the predicted probability that the habitat would be used by free-ranging pigs in relation to wild boar.
(DOCX)

## Acknowledgments

The authors would like to thank their colleagues at the EOVR-IZS (Cagliari, Sardinia), who provided data essential for carrying out this work and all those who participated in the field work and collection of data on wild boar, especially Marco Muzzeddu (Agenzia Forestas, Regione Sardegna). We also thank the EU project (H2020 VACDIVA 862874). The authors would also like to thank P. Babé for helping analyze radiocollar data.

## Author Contributions

**Conceptualization:** Jaime Bosch, Jose A. Barasona, Marcella Cherchi, Jose M. Sánchez-Vizcaíno.

**Data curation:** Jose A. Barasona.

**Formal analysis:** Jaime Bosch, Jose A. Barasona.

**Funding acquisition:** Jose M. Sánchez-Vizcaíno.

**Investigation:** Jaime Bosch, Jose A. Barasona, Estefanía Cadenas-Fernández, Cristina Jurado.

**Methodology:** Jaime Bosch, Jose A. Barasona, Estefanía Cadenas-Fernández.

**Resources:** Antonio Pintore, Daniele Denurra, Marcella Cherchi, Joaquín Vicente, Jose M. Sánchez-Vizcaíno.

**Software:** Jaime Bosch, Jose A. Barasona.

**Supervision:** Jaime Bosch, Jose A. Barasona.

**Validation:** Jaime Bosch, Jose A. Barasona.

**Visualization:** Jose A. Barasona.

**Writing – original draft:** Jaime Bosch.

**Writing – review & editing:** Jose A. Barasona, Estefanía Cadenas-Fernández, Cristina Jurado, Joaquín Vicente, Jose M. Sánchez-Vizcaíno.

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
