## [Decision Letter · Decision Letter 0]

13 Mar 2020

PONE-D-20-01666

Retrospective spatial analysis for African swine fever in endemic areas to assess interactions between susceptible host populations

PLOS ONE

Dear Dr. Bosch,

Thank you for submitting your manuscript to PLOS ONE. After careful consideration, we feel that it has merit but does not fully meet PLOS ONE’s publication criteria as it currently stands. Therefore, we invite you to submit a revised version of the manuscript that addresses the points raised during the review process.

We would appreciate receiving your revised manuscript by Apr 27 2020 11:59PM. To enhance the reproducibility of your results, we recommend that if applicable you deposit your laboratory protocols in protocols.io, where a protocol can be assigned its own identifier (DOI) such that it can be cited independently in the future. For instructions see: http://journals.plos.org/plosone/s/submission-guidelines#loc-laboratory-protocols

We look forward to receiving your revised manuscript.

Kind regards,

Douglas Gladue, Ph.D

Academic Editor

PLOS ONE

Journal Requirements:

3.  PLOS journals require authors to make all data underlying the findings described in their manuscript fully available without restriction. Therefore we ask that you please upload underlying data to an appropriate data repository and update your Data Availability Statement accordingly. More information about recommended repositories can be found here: http://journals.plos.org/plosone/s/data-availability#loc-recommended-repositories.

4. In your Methods section, please provide additional location information of the study area, including geographic coordinates for the data set if available.

5.  We note that [Figure(s) 1 and 2] in your submission contain [map/satellite] images which may be copyrighted. All PLOS content is published under the Creative Commons Attribution License (CC BY 4.0), which means that the manuscript, images, and Supporting Information files will be freely available online, and any third party is permitted to access, download, copy, distribute, and use these materials in any way, even commercially, with proper attribution. For these reasons, we cannot publish previously copyrighted maps or satellite images created using proprietary data, such as Google software (Google Maps, Street View, and Earth). For more information, see our copyright guidelines: http://journals.plos.org/plosone/s/licenses-and-copyright.

1.     You may seek permission from the original copyright holder of Figure(s) [1 and 2] to publish the content specifically under the CC BY 4.0 license.  

Additional Editor Comments (if provided):

Both reviewers agree that this manuscript could use careful proofreading. Please proofread carefully the revision before sending to avoid any delays. Please address all other comments/corrections.

Reviewers' comments:

Reviewer's Responses to Questions

**Comments to the Author**

1. Is the manuscript technically sound, and do the data support the conclusions?

Reviewer #1: Yes

Reviewer #2: Yes

2. Has the statistical analysis been performed appropriately and rigorously? 

Reviewer #1: Yes

Reviewer #2: Yes

3. Have the authors made all data underlying the findings in their manuscript fully available?

Reviewer #1: Yes

Reviewer #2: No

4. Is the manuscript presented in an intelligible fashion and written in standard English?

Reviewer #1: Yes

Reviewer #2: No

5. Review Comments to the Author

Reviewer #1: This is an interesting paper,and the results will be helpful for the control of ASF. Hence, I strongly recommend to publish this work after some minor problems:

1.The introduction should be strengthened, and some recent research of Risa assesment of ASF should be mentioned such as "Prediction for global African swine fever outbreaks based on a combination of random forest algorithms and meteorological data","Risk analysis of African swine fever in Poland based on spatio-temporal pattern and Latin hypercube sampling, 2014-2017","Assessment of the impact of forestry and leisure activities on wild boar spatial disturbance with a potential application to ASF risk of spread","Quantitative risk assessment of African swine fever virus introduction to Japan via pork products brought in air passengers' luggage","Risk assessment of African swine fever in the south-eastern countries of Europe"etc.

2.The brief introduction of methods such as Latin selection should be provided.

3.The autors should validate their model.

Reviewer #2: The manuscript presents an interesting and novel study aimed at reveling interactions between wild boar and domestic swine populations in Sardinia. Using GPS tracking data the authors demonstrate that interactions between wild boars and free-range domestic pigs tend to occur in the proximity to pig farms. Statistical metrics have been developed to quantify and map interactions. The findings may contribute into further understanding of pig diseases spread pattern in currently affected European countries.

Being not an expert in English I cannot point at the specific language issues, but it looks like the manuscript requires careful proofreading to eliminate some inaccuracies, including but not limited to listed below:

Page 4 lines 88-89: what is PPA?

Page 5 line 95: a comma is missed between Russia and Mongolia.

Page 5 lines 103-104: 'maintenance' and 'maintained' in the same sentence look not very good.

Page 9 line 194: I would suggest specifying here that a Pearson correlation coefficient was used.

Page 28 line 636: please check.

6. PLOS authors have the option to publish the peer review history of their article (what does this mean?). If published, this will include your full peer review and any attached files.

Reviewer #1: No

Reviewer #2: Yes: Fedor Korennoy

---

## [Author Response · Author response to Decision Letter 0]

29 Apr 2020

Dear Anita Estes,

Firstly, I would like to acknowledge the time spent by the referees and editor in this manuscript entitled “Retrospective spatial analysis for African swine fever in endemic areas to assess interactions between susceptible host populations”

 Answering the two points that require:

1) Please upload a copy of Figure 3 which you refer to in your text on page 16. Or if the figure is no longer to be included as part of the submission please remove all reference to it within the text.

Thank you for your comment. We delete in the text “fig.3” and we added “fig.2”.

2) Thank you for providing additional information regarding your third party data. Before we proceed, please address the following: Kindly confirm that others would be able to access these data in the same manner as the authors. Please also confirm that the authors did not have any special access privileges that others would not have

We confirm that others will be able to access this data in the same way as the authors. Also we confirm that the authors did not have any special access privileges that others would not.

Sincerely,

Yours sincerely,

Jaime Bosch and Jose Angel Barasona

##########################################################################

Dear Editor,

Thank you for the positive response on this manuscript. We have incorporated and answered all suggestions from both reviewers and feel that the manuscript has been substantially improved. On the other hand, the method section have been now improved to justify clearly this analytical approach. The introduction have been synchronized to the rest of the manuscript to permit a nicer read. According to suggestions, a final version of the entire manuscript have been edited to improve English language by a professional native speaker. Please see below the answers and comments to each reviewer. We believe that the manuscript has improved with the modifications and we hope after these amendments made the article now acceptable and suitable for publication in Plos One.

All authors have read and approved this version of the manuscript. All prevailing local, national and international regulations and conventions, and normal scientific ethical practices, have been respected. 

Sincerely,

Yours sincerely,

Jaime Bosch and Jose Angel Barasona

PLOS ONE Decision: Revision required [PONE-D-20-01666] Retrospective spatial analysis for African swine fever in endemic areas to assess interactions between susceptible host populations

1. When submitting your revision, we need you to address these additional requirements. PLOS ONE's style requirements.

The manuscript meets the style requirements of Plos One.

a) Data access restrictions. 

The occurrence records of free-range pig sightings, data from telemetry radiocollar of wild boar and ASF notifications in wild boar and domestic pigs database comes from the Istituto Zooprofilattico Sperimentale della Sardegna (EOVR-IZS) and therefore, belongs to a third-party organization and data contain potentially sensitive information. We cannot make this database publicly available, however, we provide here the contact information as requested:

Regional Veterinary Epidemiological Observatory

Istituto Zooprofilattico Sperimentale della Sardegna (EOVR-IZS)

Via Duca degli Abruzzi 8 and Via Vienna 2, 07100 Sassari (SS) - Italy

Tel. +39 079 2892200

https://www.izs-sardegna.it/

Data Availability

All variables used for spatial interspecific interaction model are publically available and their sources are detailed in material and methods, in environmental factors. 

Points 3) and 4) are related to section 2) and have already answered.

5. We note that [Figure(s) 1 and 2] in your submission contain [map/satellite] images which may be copyrighted. All PLOS content is published under the Creative Commons Attribution License (CC BY 4.0), which means that the manuscript, images, and Supporting Information files will be freely available online, and any third party is permitted to access, download, copy, distribute, and use these materials in any way, even commercially, with proper attribution. For these reasons, we cannot publish previously copyrighted maps or satellite images created using proprietary data, such as Google software (Google Maps, Street View, and Earth). For more information, see our copyright guidelines: http://journals.plos.org/plosone/s/licenses-and-copyright.

In Figure 1, the image contains the Digital Elevation Model for the study area and is an open source and comes from: USGS EROS Archive - Digital Elevation - Shuttle Radar Topography Mission (SRTM) 1 Arc-Second Global - Global Land Cover Facility (GLCF) University of Maryland, USA. And USGS is a one resource that Plos One suggests for using maps.

In figure 1 and 2, for administration map we used also a resource that Plos One suggests: Natural earth data. All versions of Natural Earth raster + vector map data found on this website are in the public domain. For the license, the data is in the public domain and can be used to carry out any project. No permission is required to use Natural Earth. Natural Earth (public domain): http://www.naturalearthdata.com
https://www.naturalearthdata.com/downloads/

Response to reviewers:

Reviewer #1: 

This is an interesting paper, and the results will be helpful for the control of ASF. Hence, I strongly recommend to publish this work after some minor problems:

We thank the reviewer for taking the time and effort to provide this revision, and for the positive appreciation of our manuscript. We greatly appreciate all the suggestions and comments made by the expert reviewer. We have carefully studied each suggestion, modified the text when needed and replied to each comment below. We certainly consider the changes proposed will contribute to improving the quality and clarity of this manuscript. Thank you.

1.The introduction should be strengthened, and some recent research of Risa assesment of ASF should be mentioned such as "Prediction for global African swine fever outbreaks based on a combination of random forest algorithms and meteorological data","Risk analysis of African swine fever in Poland based on spatio-temporal pattern and Latin hypercube sampling, 2014-2017","Assessment of the impact of forestry and leisure activities on wild boar spatial disturbance with a potential application to ASF risk of spread","Quantitative risk assessment of African swine fever virus introduction to Japan via pork products brought in air passengers' luggage","Risk assessment of African swine fever in the south-eastern countries of Europe"etc.

Thank you for your comment and for your suggestion. Following the recommendation we have now added the suggested studies and references in the introduction section, in pages 4 and 5, in lines 92-101.

 De la Torre, A., Bosch, J., Iglesias, I., Muñoz, M. J., Mur, L., Martínez‐López, B., & Sánchez‐Vizcaíno, J. M. (2015). Assessing the risk of African swine fever introduction into the European Union by wild boar. Transboundary and emerging diseases, 62(3), 272-279.

 Bosch, J., Rodríguez, A., Iglesias, I., Munoz, M. J., Jurado, C., Sánchez‐Vizcaíno, J. M., & De la Torre, A. (2017). Update on the risk of introduction of African swine fever by wild boar into disease‐free European Union countries. Transboundary and emerging diseases, 64(5), 1424-1432.

 Liang, R., Lu, Y., Qu, X., Su, Q., Li, C., Xia, S., & Niu, B. (2019). Prediction for global African swine fever outbreaks based on a combination of random forest algorithms and meteorological data. Transboundary and emerging diseases.

 Iglesias, I., Montes, F., Martínez, M., Perez, A., Gogin, A., Kolbasov, D., & de la Torre, A. (2018). Spatio-temporal kriging analysis to identify the role of wild boar in the spread of African swine fever in the Russian Federation. Spatial statistics, 28, 226-235.

 Lu, Y., Deng, X., Chen, J., Wang, J., Chen, Q., & Niu, B. (2019). Risk analysis of African swine fever in Poland based on spatio-temporal pattern and Latin hypercube sampling, 2014–2017. BMC veterinary research, 15(1), 160.

 Petit, K., Dunoyer, C., Fischer, C., Hars, J., Baubet, E., López‐Olvera, J. R., & Peroz, C. (2019). Assessment of the impact of forestry and leisure activities on wild boar spatial disturbance with a potential application to ASF risk of spread. Transboundary and emerging diseases.

 Ito, S., Jurado, C., Sánchez‐Vizcaíno, J. M., & Isoda, N. (2019). Quantitative risk assessment of African swine fever virus introduction to Japan via pork products brought in air passengers’ luggage. Transboundary and emerging diseases.

 EFSA Panel on Animal Health and Welfare (AHAW), Nielsen, S. S., Alvarez, J., Bicout, D., Calistri, P., Depner, K. & Miranda, M. A. (2019). Risk assessment of African swine fever in the south‐eastern countries of Europe. Efsa Journal, 17(11), e05861.

 Bosch J., Iglesias I., Martínez M., de la Torre A. Climatic and topographic tolerance limits of wild boar in Eurasia: implications for their expansion. GEOGRAPHY, ENVIRONMENT, SUSTAINABILITY. 2020; 13(1):107-114. https://doi.org/10.24057/2071-9388-2019-52

2. The brief introduction of methods such as Latin selection should be provided.

We agree. This introduction to the latent selection difference (LSD) function to estimate spatial interactions between different species has been now included:

“Following the approach of Barasona et al. [44], we applied latent selection difference (LSD) functions [54,55] to assess fine-scale interactions between wild boar and free-ranging pigs, as well as determine differences in the environmental factors evaluated. This analysis determined which covariates predicted similarities and differences in resource use between species at smaller spatial scales, measured by the estimated β coefficients from logistic regression [56,57,58]. Significant coefficients indicate less or more use by one species compared to other one [59,60]”

56. Hosmer DW, Lemeshow S. Applied logistic regression. Wiley; 2000. 

57. Roever CL, Boyce MS, Stenhouse GB. Grizzly bears and forestry. II: Grizzly bear habitat selection and conflicts with road placement. For Ecol Manage. Elsevier; 2008;256: 1262–1269. doi:10.1016/j.foreco.2008.06.006

58. Fischer LA, Gates CC. Competition potential between sympatric woodland caribou and wood bison in southwestern Yukon, Canada. Can J Zool. NRC Research Press Ottawa, Canada ; 2005;83: 1162–1173. doi:10.1139/z05-117

59. Latham, A. D. M., Latham, M. C., and Boyce, M. S. (2011). Habitat selection and spatial relationships of black bears (Ursus americanus) with woodland caribou (Rangifer tarandus caribou) in northeastern Alberta. Canadian Journal of Zoology 89, 267–277. doi:10.1139/z10-115

60. Latham, A. D. M., Latham, M. C., Boyce, M. S., & Boutin, S. (2013). Spatial relationships of sympatric wolves (Canis lupus) and coyotes (C. latrans) with woodland caribou (Rangifer tarandus caribou) during the calving season in a human-modified boreal landscape. Wildlife Research, 40(3), 250-260.

3.The autors should validate their model.

Thank you for this comment. We agree with the reviewer in the importance of model and data validation. For this reason, we have now clarified this suggestion in the method section (pag 12, lines 258-260). When the model was designed, we randomly split the whole datasets using 70% of locations to parameterise the models (training datasets) and the remaining 30% of locations for independent validation (validation datasets) of the training models, according to Boyce et al. Evaluating resource selection functions. Ecol Modell. Elsevier; 2002; 157: 281–300. In this sense, the same training model was fitted with the independent validation dataset (30%) to assess the predictive capacity of the model in this scenario by means of calibration plots. In this exploration, the best models were tested against the corresponding validation dataset, then the observed and predicted frequencies of observations were plotted for 10 equally sized intervals of predicted probabilities (0–1). Ideally, a LSD model with high predictive capacity should show perfectly aligned points along a 45° line, according to Pearce and Ferrier (2000). We also have assessed the ability of the model to discriminate free-ranging pig and wild boar locations by determining the area under the receiver operating characteristic curve (AUC). The best possible AUC is 1 (models with perfect discrimination ability), while a value of 0.5 suggests that the model performs no better than random (Pearce and Ferrier 2000)].In addition, in this study we have evaluated spatially and qualitatively the model and the coefficient results with the obtained biological and health data available, not used to obtain and develop the model, but used to validate the best fitting model outputs, with the ASF notifications in wild boar and domestic pigs from 2010 to 2016).

“In order to validate the outcome model, we randomly split the datasets using 70% of locations to parameterise the models (training datasets) and the remaining 30% of locations for model validation (independent, validation datasets) [52].”

“We assessed predictive capacity of the best model using calibration plots: the best models were tested against the corresponding validation dataset, then the observed and predicted frequencies of observations were plotted for 10 equally sized intervals of predicted probabilities (0–1). A model with high predictive capacity should show perfectly aligned points along a 45° line [66]. We also assessed the ability of the model to discriminate free-ranging pigs and wild boar locations by determining the area under the receiver operating characteristic curve (AUC). The best possible AUC is 1 (models with perfect discrimination ability), while a value of 0.5 suggests that the model performs no better than random [66].”

 Boyce MS, Vernier PR, Nielsen SE, Schmiegelow FK. Evaluating resource selection functions. Ecol Modell. Elsevier; 2002;157: 281–300. doi:10.1016/S0304-3800(02)00200-4

 Pearce J, Ferrier S: Evaluating the predictive performance of habitat models developed using logistic regression. Ecol Model. 2000, 133: 225-245. 10.1016/S0304-3800(00)00322-7.

Reviewer #2

The manuscript presents an interesting and novel study aimed at reveling interactions between wild boar and domestic swine populations in Sardinia. Using GPS tracking data the authors demonstrate that interactions between wild boars and free-range domestic pigs tend to occur in the proximity to pig farms. Statistical metrics have been developed to quantify and map interactions. The findings may contribute into further understanding of pig diseases spread pattern in currently affected European countries.

Dear Dr Korennoy, we are grateful for your positive comments and recommendations. We have studied and incorporated your suggestions into the text. We certainly think these changes have improved the quality of our manuscript.

Being not an expert in English I cannot point at the specific language issues, but it looks like the manuscript requires careful proofreading to eliminate some inaccuracies, including but not limited to listed below:

According to suggestions, the entire manuscript has been edited to improve the English language by a professional native speaker, Dr. Armando Chapin.

Page 4 lines 88-89: what is PPA?

Thank you for your comment. We have corrected the acronym “PPA” by “ASF” in page 4, line 91. The meaning is ASF (African Swine Fever), PPA is the acronym in Spanish: Peste Porcina Africana. We are sorry for the mistake.

Page 5 line 95: a comma is missed between Russia and Mongolia.

OK, done. line 106, page 5.

Page 5 lines 103-104: 'maintenance' and 'maintained' in the same sentence look not very good.

Thank you for the suggestion. Following the recommendation, we have replaced “maintained” by “occurred”.

Page 9 line 194: I would suggest specifying here that a Pearson correlation coefficient was used.

Thank you. We have included the Pearson correlation coefficient as suggested. We also include the reference of Pearson test that was used: Hijmans et al 2012. 

Page 28 line 636: please check.

We have checked and corrected the error in the reference (page 31, line 692). We have added issue 3 in volume 75 in the reference [75(3): 600–612]. Thank you

---

## [Editor Report · Decision Letter 1]

6 May 2020

Retrospective spatial analysis for African swine fever in endemic areas to assess interactions between susceptible host populations

PONE-D-20-01666R1

Dear Dr. Bosch,

We are pleased to inform you that your manuscript has been judged scientifically suitable for publication and will be formally accepted for publication once it complies with all outstanding technical requirements.

With kind regards,

Douglas Gladue, Ph.D

Academic Editor

PLOS ONE
---

## [Editor Report · Acceptance letter]

13 May 2020

PONE-D-20-01666R1 

Retrospective spatial analysis for African swine fever in endemic areas to assess interactions between susceptible host populations 

Dear Dr. Bosch:

I am pleased to inform you that your manuscript has been deemed suitable for publication in PLOS ONE. Congratulations! Your manuscript is now with our production department. 

With kind regards,

on behalf of

Dr. Douglas Gladue 

Academic Editor

PLOS ONE